# Development of a Conceptual Framework for Occupational Safety and Health in Palestinian Manufacturing Industries

**DOI:** 10.3390/ijerph18031338

**Published:** 2021-02-02

**Authors:** Hanan S. Tuhul, Amer El-Hamouz, A. Rasem Hasan, Hanan A. Jafar

**Affiliations:** 1Hayara Consulting Engineering Office, P.O. Box 10, Tulkarm, Palestine; eng_h_tuhul@hotmail.com; 2Chemical Engineering Department, An-Najah National University, P.O. Box 7, Nablus, Palestine; 3Civil Engineering Department, An-Najah National University, P.O. Box 7, Nablus, Palestine; h.jafar@najah.edu; 4Research and Development Department, Water and Environmental Studies Institute, An-Najah National University, P.O. Box 7, Nablus, Palestine

**Keywords:** conceptual framework, Occupational Safety and Health, occupational accidents, occupational injuries, OSH rates, frequency severity index, safety performance factor, syndicates

## Abstract

The annual increase in the number of occupational accidents and diseases in the Palestinian manufacturing industries confirms a serious problem that threatens the Occupational Safety and Health (OSH) in such industries, with negative consequences in the society and economy. As the Palestinian OSH data are insufficient, tightwad, and with discrepancies in published data by different agencies, this study aimed to investigate the OSH status in the Palestinian manufacturing industries and determine the Palestinian OSH trends rates based on international practice in the EU and USA. Also, to shed light on the OSH situation in the manufacturing sector and warrant the inspection and monitoring of industries by the respective officials. The OSH data of 175 industrial establishments and 199 industrial employees was collected by meetings, interviews, and structured questionnaires for the period 2009–2016. The US and EU OSHA (Occupational Safety and Health Administration) formulas were used to calculate the OSH rates. The analysis showed that 32.30% of the industrial employees suffered from occupational injuries. The average injury rate was 7566 per 100,000 workers, indicating a low OSH level in the Palestinian manufacturing industries. The leather industry was found as the most dangerous in terms of OSH, with an average safety performance factor (SPF) of 145.6 off days/accident. Pearson’s chi-square test (χ2) revealed a correlation between occupational accidents and injuries and the employees’ commitment and lost working days. An OSH framework was developed based on safety and sustainable development pillars to assure effective enforcement of the OSH law and prevent future occupational accidents and diseases.

## 1. Introduction

Despite the increasing interest in Occupational Safety and Health (OSH), about 6300 deaths and 860,000 nonfatal occupational injuries are recorded daily worldwide [1,2,3,4,5,6,7]. Apart from threatening the life and health of workers [8,9,10,11], workplace accidents are highly sound economic sense [12,13]. Businesses spend $170 billion/year on costs associated with occupational injuries and illnesses [14,15], and about 4.0% of the world’s Gross Domestic Product (GDP) is swallowed up by the direct and indirect costs of occupational accidents and diseases [16,17].

Human behavior and the industry nature are significant drivers for the occurrence and re-occurrence of occupational accidents [18,19,20]. While there have been substantial reductions in accidents resulting from technological failures, industrial accidents due to human error have significantly increased, representing a contribution of up to 80% [21,22]. However, this increase in the percentage might be attributed to the rate-reducing other causes [23,24], and thus warrant investigations.

Accurate statistics and figures on the number of occupational accidents or fatalities in the Middle East (ME) are minimal [25,26], specifically in the manufacturing, mining, and construction industries [27,28,29,30]. For instance, Gulf Cooperation Countries (GCC) have a large expatriate population whose majority originate from India, Pakistan, Nepal, and other Asian countries suffer from occupational accidents [31,32]. For the past few years, labor advocacy groups in Qatar have raised concerns about the increasing numbers of deaths and disabilities resulting from work-related injuries [33].

In Palestine, the occupational accidents (OA) statistics are published by the Ministry of Labor (MoL) and the Ministry of Health (MoH) through their annual reports and documented as well by the Palestinian insurance companies (PICs) and the General Federation of Palestinian Trade Unions (GFPTU) [34,35]. Of particular interest among different OA types are the accidents occurred in the Palestinian manufacturing industries, hereafter called the manufacturing accidents (MA). 

The updated total OA statistics in Palestine with all subcategories are given in Table 1 [36], where the manufacturing sector is the largest contributor to these statistics. The MoL and other governmental and non-governmental institutions in Palestine try to improve the OSH conditions inside facilities, mainly for the manufacturing industries [37,38,39], due to their higher workforce and impact on the Palestinian economy, among other sectors [38]. 

The high number of occupational injuries and illnesses documented by the Palestinian labor law (PLL) indicate the Palestinian manufacturing industries’ critical situation and the crucial need for OSH measures [40]. 

Previous studies such as Al Habeel and Aiesh, 2012, Mohammed, 2014, and Abu Zeiter, 2018 [41,42,43] presented the OSH procedures in general, while others such as Al Moghny, 2006, and Salem, 2009 [44,45] discussed the OSH in the manufacturing sector in Gaza strip and Tulkarm city respectively, but without development of any system, model, or framework to solve the related safety problems. Recently, two studies [39,46] by the international labor organization (ILO) were conducted on several Palestinian workplaces and concluded the need to study the OSH problem and work-related accidents from different aspects, like non-compliance with laws, lack of emphasis on sanctions, and the absence of essential laws and regulations, to understand their causes and develop preventative measures to mitigate their dangerous consequences. 

According to the United States Agency for International Development (USAID) and the Palestinian Federation of Industries (PFI) [47], the Palestinian industrial sectors employ about 13.0% of the total workforce and contribute 16.0% to the GDP. These sectors constitute 57.1% of the whole manufacturing sector in the 11 governorates of the West Bank (WB) part of Palestine. They include more than 57.0% of the workforce [48], which endorses the need to highlight OSH conditions in these sectors. In this context, the presented study aimed to investigate the Palestinian OSH trends and associated rates for the Palestinian manufacturing sector’s industrial establishments. Additionally, this research aimed to develop an OSH framework that governs the OSH situation in Palestine.

## 2. Materials and Methods

### 2.1. Data Collection and Sample

This research has targeted eight Palestinian industrial sectors with a significant weight in the Palestinian economy: metal and aluminum, paper and cartoon, leather and shoes, food and beverages, chemical, wood and furniture, and plastic industries.

The targeted industries in this research are characterized as statistically heterogeneous populations due to their differences in size, the number of employees, and contribution to the GDP. There was a need to determine the sample size (number of establishments) per each industrial sector and in each governorate to have a representative sample. The stratified multistage sampling technique and Thompson Formula [49,50] were used to determine the sample size. In such a method, the whole heterogeneous population can be divided into smaller groups concerning the study’s characteristics. Also, it allows for three or more stages of sampling units, which was required for this research [51,52]. 

Accidents’ data with consequences were collected from the MoL and MoH’s annual reports and collected directly from PICs and GFPTU. Meetings and interviews were held with the OSH stakeholders to explore their opinions concerning the OSH and discuss their cooperative role in the improvement process. Additionally, two structured questionnaires were developed and reviewed by experts from the academia: the workplace questionnaire was administered to the establishments’ owners or managers, hereafter called employers as well, and the employees’ questionnaire was administered to the establishments’ employees. 

Out of the total 8948 registered industrial establishments in Palestine, 265 establishments were categorized under the 8 targeted manufacturing sectors and located in the West Bank (WB) 11 governorates, and 175 establishments responded to the workplace questionnaire. Later, a more representative sample of 102 industrial establishments was considered for further statistical analysis. Some of the industries either closed their business or changed their activities during the period of concern. 

In terms of the selected establishments’ employees, 199 employees were selected and interviewed. 

### 2.2. Data Analysis

#### 2.2.1. Thematic Analysis Approach

Seven semi-structured interviews were conducted with officials from six different OSH stakeholders (MoL, Civil Defense (CD), Palestinian Insurance Federation (PIF), PICs, Palestinian Capital Market Authority (PCMA), and General Federation of Palestinian Trade Unions (PGFTU)) to investigate the OSH status in the Palestinian industrial sector. According to Braun and Clarke [53], interviews were transcribed and analyzed using the thematic analysis approach. 

#### 2.2.2. Statistical Analysis

The quantitative data from the questionnaires were analyzed using Statistical Package for Social Sciences, IBM SPSS Statistics for Windows, Version 26.0 (IBMD Corp., Armonk, NY, USA), to illustrate the collected data, and test for possible correlations between the number of occupational accidents and both the human effect and the nature of industry. 

#### 2.2.3. OSHA Rate Formulas

The US and EU OSHA formulas were used to calculate the OSH rates (Table 2).

US-OSHA Formulas:OSHA Incident Rate based on injuries and illnesses (IR1) [54]:(1)$$IR1\;=\;\frac{\mathrm{Number}\;\mathrm{of}\;\mathrm{injuries}\;\mathrm{and}\;\mathrm{illnesses}}{\mathrm{Totally}\;\mathrm{worked}\;\mathrm{labor}\;\mathrm{hours}}\;\times 200,000$$OSHA Incident Rate based on lost workdays (IR2) [54]:(2)$$IR2\;=\;\frac{\mathrm{Number}\;\mathrm{of}\;\mathrm{lost}\;\mathrm{workdays}}{\mathrm{Totally}\;\mathrm{worked}\;\mathrm{labor}\;\mathrm{hours}\;}\;\times 200,000$$Fatal Accident Rate (FAR) [54]:(3)$$FAR\;=\;\frac{\mathrm{Number}\;\mathrm{of}\;\mathrm{Fatalities}}{\mathrm{Totally}\;\mathrm{worked}\;\mathrm{labor}\;\mathrm{hours}}\;\times 10^{8}$$Lost Time Case Rate (LTC) [55]:(4)$$LTC\;=\;\frac{\mathrm{Number}\;\mathrm{of}\;\mathrm{lost}\;\mathrm{Time}\;\mathrm{cases}}{\mathrm{Totally}\;\mathrm{worked}\;\mathrm{labor}\;\mathrm{hours}\;}\;\times 200,000$$Severity Rate (SR) [56]:(5)$$SR\;=\;\frac{\mathrm{Total}\;\mathrm{number}\;\mathrm{of}\;\mathrm{lost}\;\mathrm{workdays}}{\mathrm{Total}\;\mathrm{number}\;\mathrm{of}\;\mathrm{recordable}\;\mathrm{incidents}}$$

EU-OSHA Formulas:Fatality Rate (FR) [54]:(6)$$\mathrm{FR}\;=\;\frac{\mathrm{Number}\;\mathrm{of}\;\mathrm{Fatalities}\;\mathrm{per}\;\mathrm{Year}}{\mathrm{Total}\;\mathrm{number}\;\mathrm{of}\;\mathrm{Employees}}\;$$Workplace Injury Rate (IR) [57]:(7)$$\mathrm{IR}\;=\;\frac{\mathrm{Number}\;\mathrm{of}\;\mathrm{Fatal}\;\&\;\mathrm{Non}-\mathrm{Fatal}\;\mathrm{injuries}}{\mathrm{Total}\;\mathrm{number}\;\mathrm{of}\;\mathrm{Employees}}\;\times 100,000\;$$Accident Frequency Rate (AFR) [57]:(8)$$\mathrm{AFR}\;=\;\frac{\mathrm{Number}\;\mathrm{of}\;\mathrm{workplace}\;\mathrm{accidents}}{\mathrm{Totally}\;\mathrm{worked}\;\mathrm{labor}\;\mathrm{hours}}\;\times 1000,000$$Accident Severity Rate (ASR) [57]:(9)$$\mathrm{ASR}\;=\;\frac{\mathrm{Number}\;\mathrm{of}\;\mathrm{workdays}\;\mathrm{lost}\;\mathrm{to}\;\mathrm{accidents}}{\mathrm{Totally}\;\mathrm{worked}\;\mathrm{labor}\;\mathrm{hours}}\;\times 1000,000$$Occupational Disease Incidents Rate (ODIR) [57]:(10)$$\mathrm{ODIR}\;=\;\frac{\mathrm{Number}\;\mathrm{of}\;\mathrm{occupational}\;\mathrm{diseases}\;\mathrm{cases}}{\mathrm{Total}\;\mathrm{number}\;\mathrm{of}\;\mathrm{Employees}}\;\times 100,000\;$$

In both formulas, the Total Worked Labor Hours (TWLH) was calculated depending on the information and equations below:

Total Worked labor Hours [54]:(11)TWLH = H×D×E 
where (H) is the number of working hours per day = 8 h / day, 

(D) is number of working days per year = 276 Days / year, 

(E) is the total number of employees  = 2208 h/year.

In addition, the SPF was calculated based on Reference [58]:(12)$$\mathrm{SPF}\;=\;\frac{\mathrm{Total}\;\mathrm{working}\;\mathrm{hours}\;\mathrm{lost}}{\mathrm{Total}\;\mathrm{number}\;\mathrm{of}\;\mathrm{incidents}}\;$$
and the Frequency Severity Index (SFI) was calculated according to Reference [56]:(13)$$\mathrm{FSI}\;=\;\frac{\sqrt{\left(AFR\times ASR\right)}}{1000}$$

The severity and performance indices were applied to determine the most severe industrial sector.

#### 2.2.4. OSH Correlation

Pearson’s chi-squared test (χ^2^) with a significance level (α) ≤ 0.05 was applied to find the correlations between occupational accidents/injuries and the different OSH variables relative to the industrial establishments and their employees from one side, and the effect of both the human factor and the industry nature from the other side. Based on the study results, an OSH framework that governs the OSH situation is to be developed and consulted with stakeholders. 

## 3. Results

### 3.1. Collected Data and Interviews’ Analyses

A tremendous discrepancy has been noticed between the occupational accidents statistics documented by the Ministry of Labor (MoL) and those reported by the Palestinian insurance companies (PICs) [34,35], with the documented cases by the PICs being of 5 to 6 times higher than those published by the MoL (Figure 1 and Figure 2). There is a need to more precisely determine the associated OSH rates as a first step for the sustainable development of the industrial sector in Palestine. 

The interviews’ analysis and the developed central themes are shown in Figure 3. The interviewees uncovered two contradictory groups of factors (Figure 4): the first group contains the obstacles that impede the development of the OSH in the industrial sector, while the other clarifies the most influential factors needed during the OSH development process. 

### 3.2. Questionnaires’Analysis

#### 3.2.1. Survey Population

Workplace Survey

About 41.0% of the responding establishments were established after the official enforcement of the PLL. Metal industries were at the top of the respondents’ list with 23.3%. In terms of educational qualification, 54.9% of the respondents have bachelor’s degrees and more, while 29.5% have secondary certificates or less. 

Employees’ Survey

The majority of employees were males, with 93.9% of the employees’ questionnaire respondents. Youths have a strong presence in this sector as 59.1% of the respondents were less than 30 years old. Regarding the working experience, 41.5% of the respondents had an estimated working experience of 5 years or less. 

#### 3.2.2. OSH Requirements and Tools

Table 3 shows the recent percentages of the Basic Health and Safety Requirements (BHSR) availability in industrial establishments.

The analysis showed an obvious variation in OSH tools’ availability in industrial establishments, as illustrated in Figure 5.

#### 3.2.3. Exposure to Risk

Regarding the OSH in industrial sectors, about 2913 industrial employees were exposed to occupational risks, 35.0% were under 26 years old. Furthermore, 69.6% of the exposed employees had secondary education certificates or less, while 10.4% were completely illiterate. About 15.0% of these employees had 12 months or less work experience.

In terms of employment contracts, 45.1% of the exposed employees’ contracts were temporary contracts. Besides, 40.0% of the exposed employees’ contracts were not officially documented. The highest percentages of exposed employees were reported in the plastic sector with 28.0%, while the lowest rate was in the leather industrial sector with 4.0%.

#### 3.2.4. Occupational Accidents and Diseases

During 2017, about 81.0% of the industrial establishments experienced at least one occupational accident case. The highest reported percentage was in the industrial food sector, where 32.3% of the employees suffered from occupational injuries. A considerable annual upward trend has been noticed during 2009–2016 in occupational accidents, injuries, fatalities, and diseases (Figure 6). 

Related to occupational diseases, 32.4–33.5% of the employees had preliminary examinations, and 25.8–31.4% had periodic inspections. Approximately 9.1% of the industrial employees suffered from chronic occupational diseases. According to employees, the highest occupational diseases’ percentage was recorded in metal industries, whereas from employers’ perspective, the highest rate was reported in food industries (Figure 7). About 49.4% of the industrial establishments at the workplace level had at least one occupational disease case during 2017. 

From employees’ perspective, the most common occupational disease was joints disease (53.3%), while employers reported that joints diseases and respiratory problems were the most common (Figure 8).

#### 3.2.5. Accidents’ Causes

The results showed that most employers and employees refer to the causes of the recurrence of occupational accidents to the carelessness of the employees. However, 36.8% of the employers think that the non-use of the OSH personal tools was the main reason.

#### 3.2.6. Lost Working Days

Due to these occupational accidents in the manufacturing sector, the absence average was 60.2 days/accident. The highest average of lost working days due to accidents was in metal and aluminum industries, with an average of 146.5 days/accident, whereas the lowest average was recorded in leather and shoe industries (14.3 days/accident).

#### 3.2.7. Financial Losses

The industrial establishments have incurred substantial financial losses, with the highest losses in employees’ compensations (Figure 9). 

#### 3.2.8. External Investigation

The workplace results revealed that competent official OSH bodies investigated 89.9% of the industrial establishments. Table 4 shows the official entities responsible for these investigation processes according to employers’ and employees’ answers.

#### 3.2.9. Environmental Considerations

According to the workplace survey, 73.3% of the industrial establishments produce industrial wastes. The primary industrial wastes disposal method was waste dumps, followed by recycling and selling (Figure 10). 

### 3.3. OSH Trends and Rates in the Palestinian Manufacturing Sector

The total number of documented work-related injuries has fluctuated during 2009–2016, as shown in Table 5.

The metal and plastic industries sectors witnessed the highest number of occupational accidents among the manufacturing industries in Palestine (Figure 11), which is in line with other national and international studies [59,60,61]. 

A summary of the OSH statistics and estimated financial losses of each industrial sector under study is shown in Table 6. 

Based on data presented in Table 6, the US/EU OSHA rates were calculated and summarized in Table 7 and Table 8. It is concluded from these data that around 68.0% of accidents ended in injuries and/or deaths. 

### 3.4. Incident and Fatality Accident Rates

The average number of days lost per each occupational injury/disease is 2.8 days, and for every 100 full-time employees (Table 7), 10.3 workers are involved in occupational accidents/diseases compared to 20 and 3.5 for Ireland and the USA, respectively [62,63].

Plastic and metal industries are of the most dangerous sectors due to their high IR1 and IR2. In the chemical and leather sectors, for every 100 full-time employees, about 86.4% and 95.4% of employees respectively, suffer lost time due to occupational injury/disease. This is in line with UK results presented by the UK Health and Safety Executive (HSE) [64].

The average IR value presented in Table 8 and Figure 12 rings an alarm for Palestine’s severe safety situation compared to other countries’ IR values.

### 3.5. Occupational Incident and Accident Frequency Rates

The ODIR values showed an increasing trend behavior, while IR values were slightly decreasingly fluctuating and eventually decreased to around 6200 (Figure 13). On the other hand, the AFR values were nearly constant over the study period, whereas ASR doubled over the same period (Figure 14).

A more in-depth investigation of the calculated injury rates based on US-OSHA equation rates for each industrial sector is shown in Figure 15, and in accordance with Lagerstrom et al., 2019, and Yanar et al., 2019, on the danger of both the metal and wooden industries all over the world [24,65]. The industrial food sector has the highest number of occupational diseases, the second-highest number of occupational accidents, and the highest number of occupational fatalities during the scoping period, leading to a tangible value of IR1 above the average IR1. Metal and wood industries have an IR1 value above average, but after years it declines (Figure 15). These results show that the decreasing trend of IR1 values does not mean that the workplace is safe, as ODIR shows the opposite behavior.

### 3.6. Frequency Severity Index and Safety Performance Factor

Due to the limited availability of workplace-related safety data, frequency severity index (FSI) and SPF were calculated based on work time lost due to occupational incidents during 2009–2016. 

Metal industries had the highest average FSI of 3.79, while chemical industries had the lowest average FSI value of 0.54 (Figure 16). Figure 17 shows the FSI values of the entire industrial sector over the period 2009–2016.

The increasing trend in the SPF indicator (Figure 18) sheds light on the industrial manufacturing sector’s severity. Leather industries had the highest number of lost working days, with an average SPF of 145.6 working days lost/occupational accident. On the other hand, plastic industries had the lowest average SPF of 9.8, although it has the second highest FSI.

Figure 19 classifies the Palestinian industrial sectors according to their seriousness, starting with the sector with the highest SPF and ending with the lowest SPF. The figure also shows the classification variations once FSI is used as a comparison indicator.

### 3.7. OSH Correlations

The correlation hypotheses for all cases are illustrated in Appendix A. Case correlation (1) confirmed that while providing OSH requirements and developing OSH strategies affect occupational disease occurrences only, the employees’ commitment follow-up has its apparent effects on the numbers of the occurring occupational accidents/injuries/diseases. Simultaneously, these occupational accidents/injuries and diseases impact the number of lost working days/hours (Figure 20). According to case correlation (2), wherever the industrial establishments are, or to what sector they belong, this does not affect their orientation or attitudes towards their industries, employees, and all the OSH issues (Figure 21). 

Case correlation (3) is another illustration of correlations between the OSH variables. The correlation revealed that employees of ages 21–25 years old were the most prone to occupational injuries, while those of ages 26–30 years old were the most likely to suffer from occupational diseases.

While age was the only independent variable that correlates with employees’ injuries, employees’ diseases were affected by other variables. The analysis showed that employees with more than 10 years of experience were suffering from occupational diseases (Figure 22).

In terms of occupational diseases, the problem seems to be much more complicated. There is a non-existent correlation according to case (4) between the primary and periodic medical examinations, that are considered the most essential and necessary OSH requirement, and the occupational diseases mainly originate due to the failure to conduct these examinations as one of the Labor Law (PLL) violations (Figure 23).

## 4. Discussion

There are neither definite OSH objectives nor belief in the Palestinian OSH system. What is more noticeable is the low OSH awareness. This poor awareness is not limited to one category; on the contrary, it is a severe societal problem. However, this can be considered an international problem, as Jedynska et al. clarified in their report to the European Agency for Safety and Health at Work (EU-OSHA) [66]. 

The kernel of an effective safety system is a perfectly designed and planned system, providing that raising safety awareness is a priority, developing safety strategies is a must, and the existence of safety requirements and tools is a fact supported by every possible affecting safety stakeholder (Figure 24). Nevertheless, this kernel cannot be formed or shaped unless there is a severe involvement, commitment, responsibility, and cooperation envelope created between every involved safety stakeholder, from each senior official to the most junior employee [33]. Finally, a commitment envelope will never be efficient without being guided by the law and regulations developed and supervised by the main safety officials (MoL, CD), as approved in case correlation (1). This case of correlation had enforced the importance of the OSH strategies and OSH requirements tools for more safe workplaces. Moreover, case correlation (2) promoted the reality of the different OSH stakeholders’ negative attitude towards the OSH issues, assuring the absence of external investigations, awareness activities, practical cooperation, and real desire to understand the OSH fundamentals and recognition the OSH problem. Yanar et al. proved in their study that the supportive investigation in the workplace besides supervisors could play an essential role in creating a safe work environment and reducing the risk of injury among workers [24]. On the other hand, case correlation (3) reflected the consequences of the lack of OSH awareness, the employees’ reckless attitude, and the employee’s stability. Moreover, the non-existent correlations that appeared in the case correlation (4) between the primary and periodic medical examinations mainly originate from the failure to conduct these examinations as one of the Labor Law (PLL) violations. Other significant factors that affect the OSH status in Palestine were the law, government, and the resistance to change. Despite all the strenuous efforts and hard work exerted by the MoL in improving the Palestinian OSH reality, real tangible deficiencies existed concerning OSH law enforcement policies and methodologies.

To overcome OSH’s previous issues in Palestine, a conceptual OSH framework has been developed (Figure 25 and Figure 26). Countries worldwide have recognized the necessity of developing different strategies, systems, and frameworks to mitigate the occurrences of occupational accidents [33,67,68,69,70]. Based on this research and consultation with stakeholders, the outstanding framework was divided into four main phases. The first phase (government phase) is related to OSH governmental commitment and national support. This phase represents the charging and stimulation step for the leading interested authorities toward a practical OSH project. At the same time, it keeps supporting the national industries and their products to give their best. The last step to ensure promoting industries and backing authorities is the needed financial support. 

In the second phase, the entities in the officials’ phase derive the financial, moral, and national support from phase one to promote their commitment, responsibility, involvement, coordination, and indeed feedback and documentation toward the government, the project, the stakeholders and employees, and the society. This phase witnesses the coordination, planning, individuals’ development, and cooperation with the industrial sectors. For the sake of better results, plans, preparations, and research should be promoted in this phase and the next phases.

The third phase is the participation phase, where each stakeholder starts to mark and contribute to the project. This phase’s start is the awareness stage, developing the perception toward the OSH issues, training courses, promotions to identify the workplace hazards, and occupational accidents and diseases, in addition to strategic planning, building an information base, reorganizing the establishments’ systems, determining OSH requirements, promoting positive attitudes, strengthening responsibilities and involvement, and offering motivational incentives for stakeholders, employees’ cooperation, and feedback. 

The last phase is the program development phase, in which the system will be developed, tested, evaluated, and operated to ensure sustainable development. Each stage will be accompanied by revisions, feedback from each input, monitoring for outputs, and documentation for all observations, findings, strengths and weaknesses, failure, and success. 

These phases’ success depends on many factors: the continuous work, joint efforts, and interrelated activities, where all are to be conducted and executed within a thoughtful timeframe. Besides, a well-designed action plan that starts with the government’s belief and commitment and lasts as long as a real outstanding OSH framework is prioritized. Moreover, it is crucial to take into consideration all measures and procedures that are strongly needed to ensure:
Successful application and enforcement of all the framework phases.Rigorous monitoring and following up on the framework development process away from any possible work obstacles that may result from issues like corruption, disruption, procrastination, indifference, or even any intentional and unintentional errors.

A qualified in-charge supervisory body was included in the framework’s structure and assumed the primary responsibility of overseeing and advocating the OSH framework’s development and its technical, financial, and temporal requirements. Figure 25 illustrates the structure of the proposed OSH syndicates and Figure 26 represents the OSH outstanding conceptual framework for the Palestinian industrial sectors.

## Figures and Tables

**Figure 1 ijerph-18-01338-f001:**
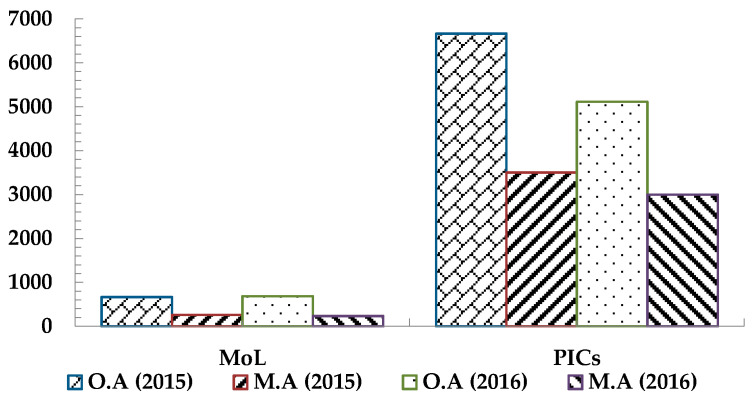
Number of manufacturing accidents and the total occupational accidents according to the Ministry of Labor (MoL) and the Palestinian insurance companies (PICs) reports (2015/2016), where O.A = Occupational Accidents, M.A = Manufacturing Accidents.

**Figure 2 ijerph-18-01338-f002:**
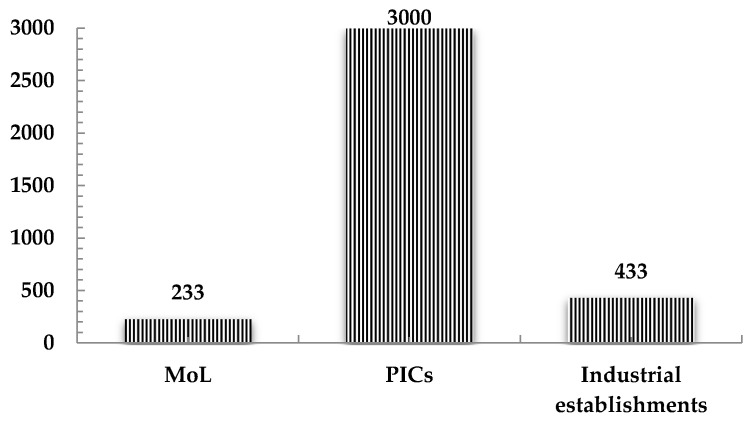
The number of occupational accidents (O.A) according to the Ministry of Labor (MoL) and Palestinian insurance companies (PICs) (2016–2017) reports [35,36,37,38,39,40], and those obtained from the industrial establishments during this research’s survey.

**Figure 3 ijerph-18-01338-f003:**
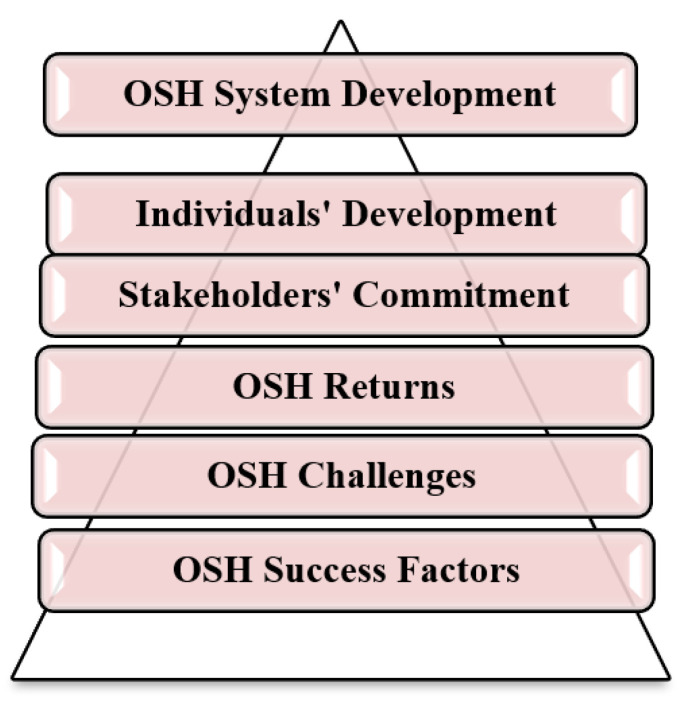
The central themes developed based on the conducted interviews.

**Figure 4 ijerph-18-01338-f004:**
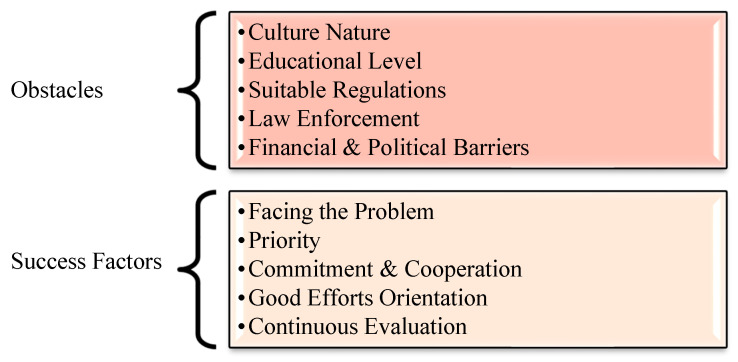
The obstacles and influences in the OSH development process.

**Figure 5 ijerph-18-01338-f005:**
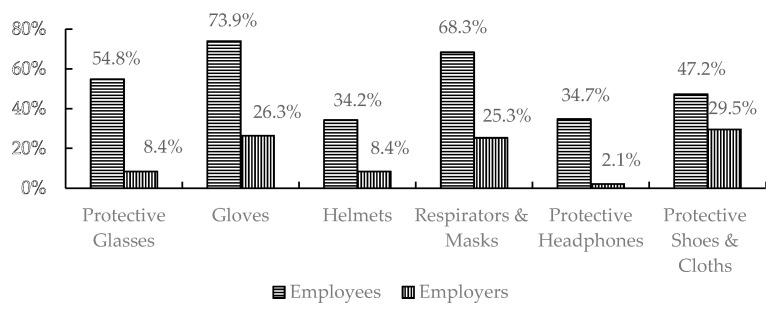
Percentages of OSH tools’ existence according to both employees’ and workplace survey analysis.

**Figure 6 ijerph-18-01338-f006:**
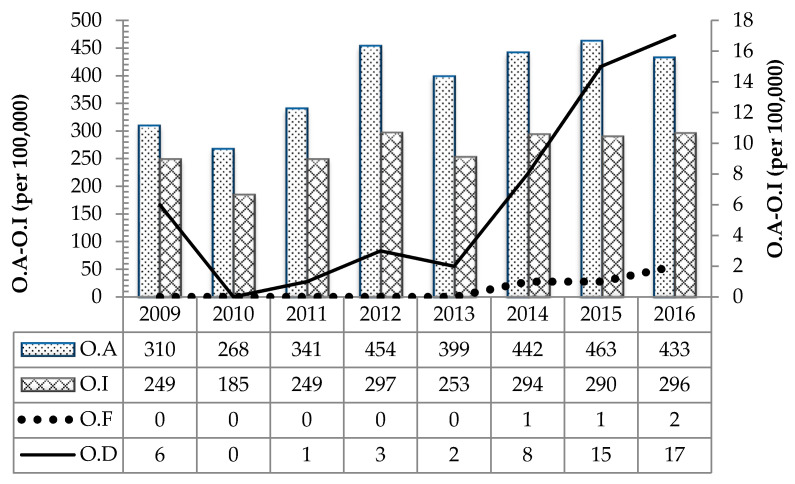
The OSH statistics in the industrial sectors between the years 2009 and 2016. Where: O.I = Occupational Injuries, O.A: Occupational Accidents, Occupational Diseases, and O.F: Occupational Fatalities.

**Figure 7 ijerph-18-01338-f007:**
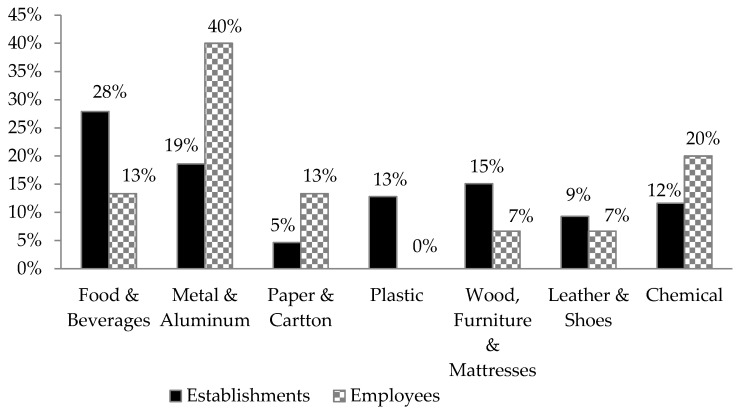
The occupational diseases’ percentages per industrial sector according to the employees and their employers.

**Figure 8 ijerph-18-01338-f008:**
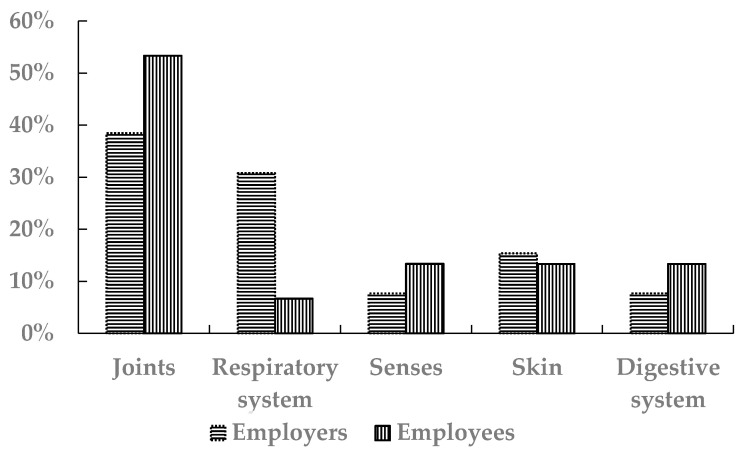
The percentages of the types of the occupational disease in the industrial sectors for the employees and the employers.

**Figure 9 ijerph-18-01338-f009:**
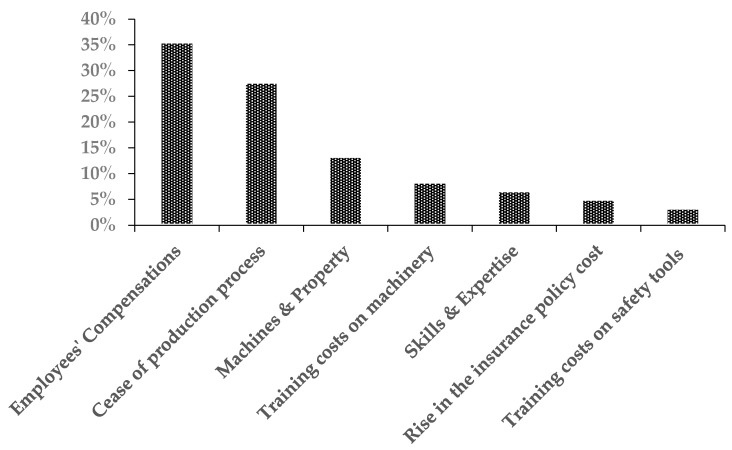
The percentages of the industrial establishments’ losses due to occupational accidents.

**Figure 10 ijerph-18-01338-f010:**
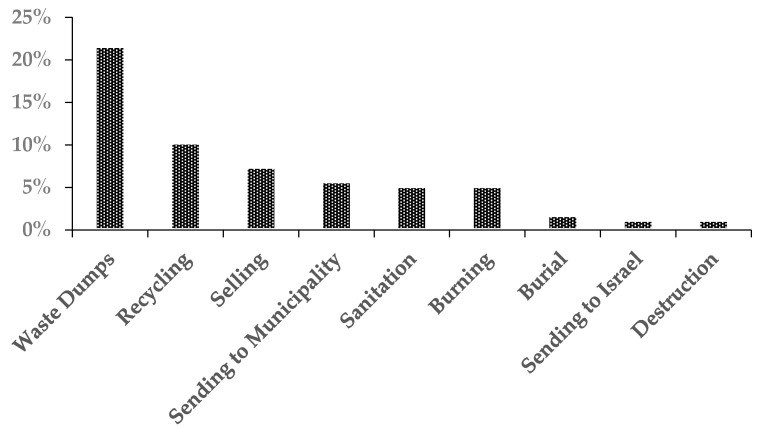
Percentages of industrial waste disposal methods.

**Figure 11 ijerph-18-01338-f011:**
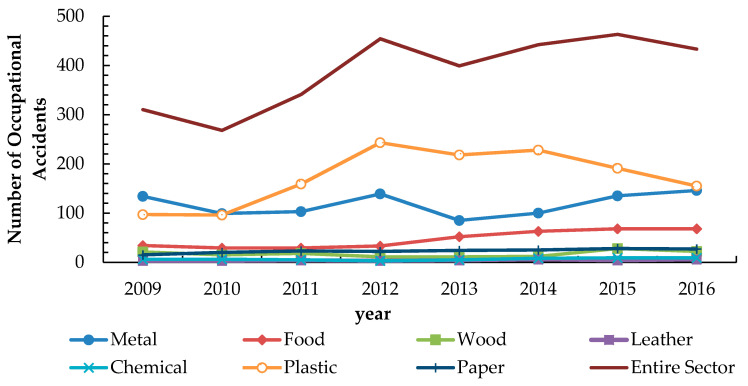
Occupational accidents in all Palestinian industrial sectors during 2009–2016.

**Figure 12 ijerph-18-01338-f012:**
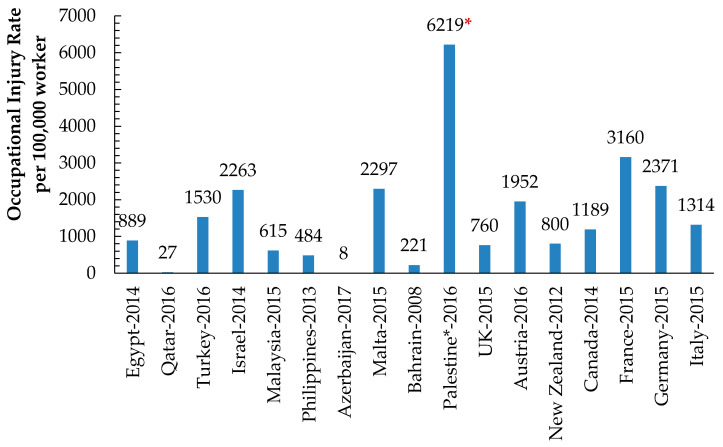
The Occupational Injury Rate (per 100,000 workers) in Palestinian manufacturing industries compared to those of other international countries. * The Calculated IR of 2016 obtained OSH statistics for the Palestinian industrial establishments.

**Figure 13 ijerph-18-01338-f013:**
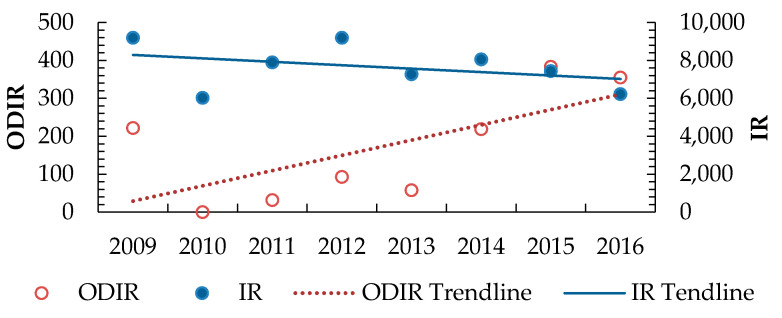
ODIR and IR of the industrial sectors during 2009–2016. Where ODIR = Number of workplace diseases of the total number of employees per 100,000 employed persons, IR = Number of workplace injuries of the total number of employees per 100,000 employed persons.

**Figure 14 ijerph-18-01338-f014:**
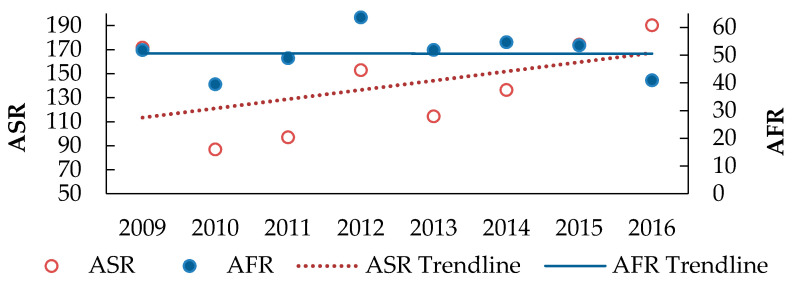
ASR and AFR Rates of the industrial sectors during 2009–2016. Where ASR = Number of reported human-days lost × 1,000,000/number of human-hours worked, AFR = Number of workplace accidents reported per number of human-hours worked × 1,000,000.

**Figure 15 ijerph-18-01338-f015:**
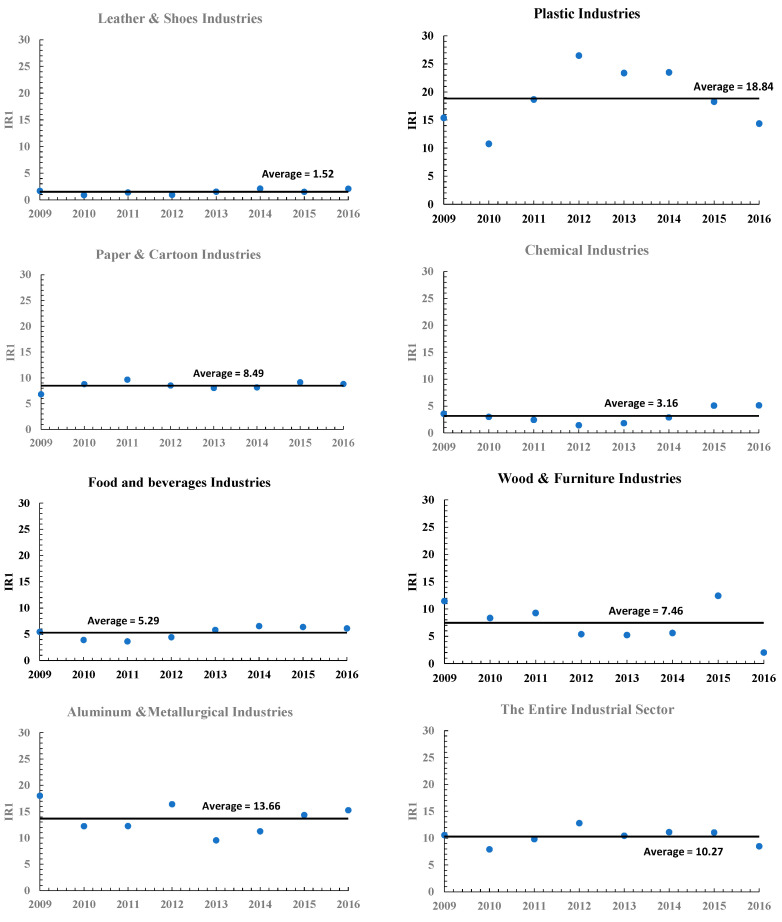
The Occupational Safety and Health Administration (OSHA) Incident Rate, based on injuries and illnesses (IR1) rates (per 100 full-time workers) for each targeted industrial sector during 2009–2016.

**Figure 16 ijerph-18-01338-f016:**
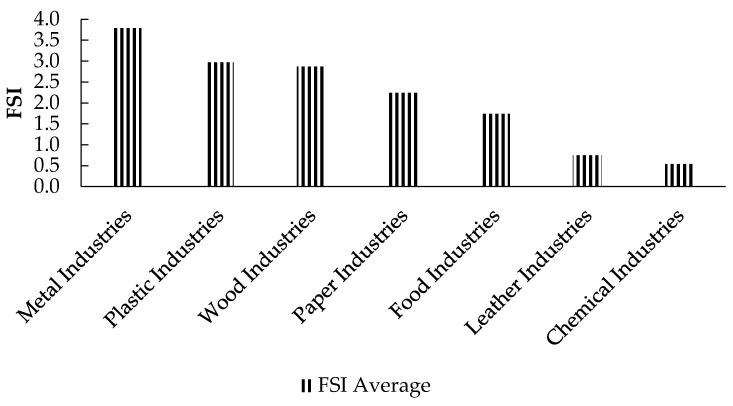
The severity classification of the Palestinian industrial sectors according to the FSI indicator.

**Figure 17 ijerph-18-01338-f017:**
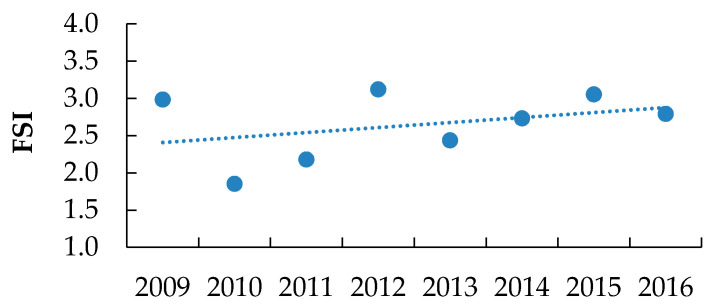
The severity classification of the Palestinian industrial sectors according to the FSI indicator during 2009–2016.

**Figure 18 ijerph-18-01338-f018:**
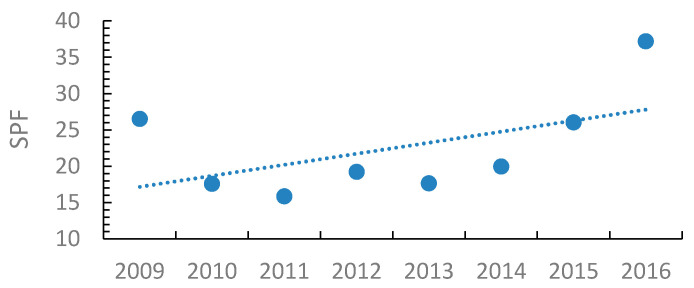
The SPF indicators for the Palestinian industrial sector between the years 2009 and 2016, where SPF = Number of working hours/days lost due to each occupational accident, regardless of its severity.

**Figure 19 ijerph-18-01338-f019:**
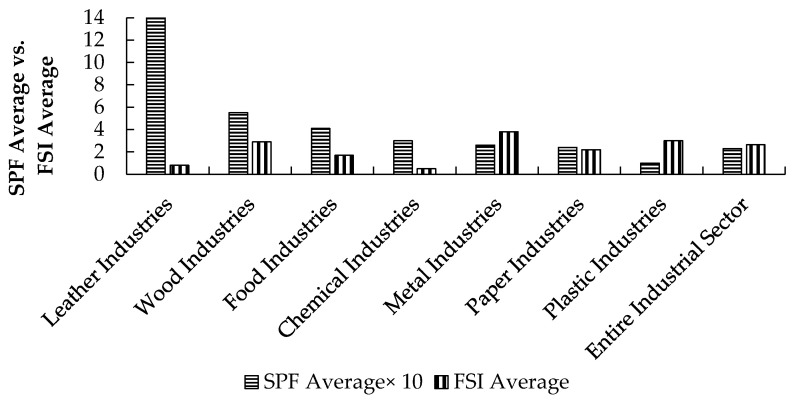
Severity classification of the Palestinian industrial sectors according to SPF and FSI indicators.

**Figure 20 ijerph-18-01338-f020:**
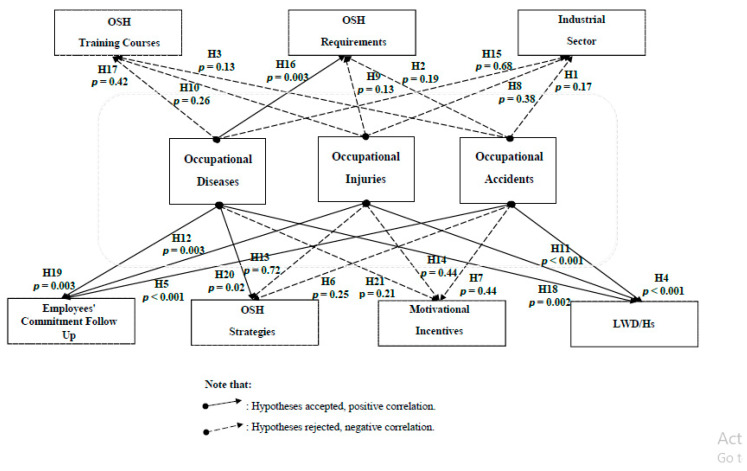
The presentation of case correlation (1) and its hypothesis testing. LWD/h is the loss of working days or hours.

**Figure 21 ijerph-18-01338-f021:**
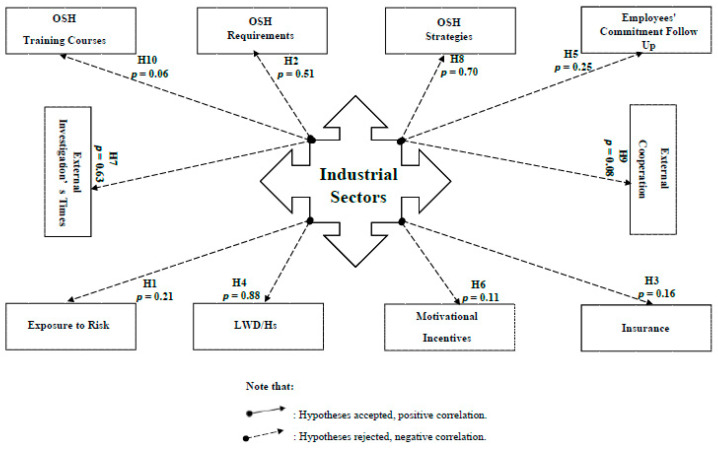
The presentation of case correlation (2) and its hypothesis testing.

**Figure 22 ijerph-18-01338-f022:**
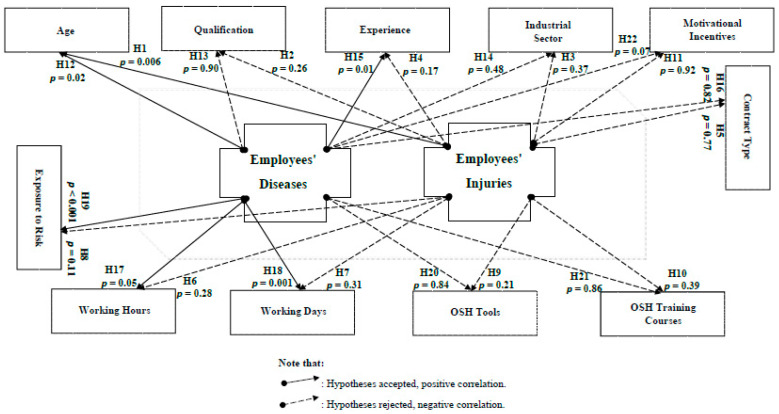
The presentation of case correlation (3) and its hypothesis testing.

**Figure 23 ijerph-18-01338-f023:**
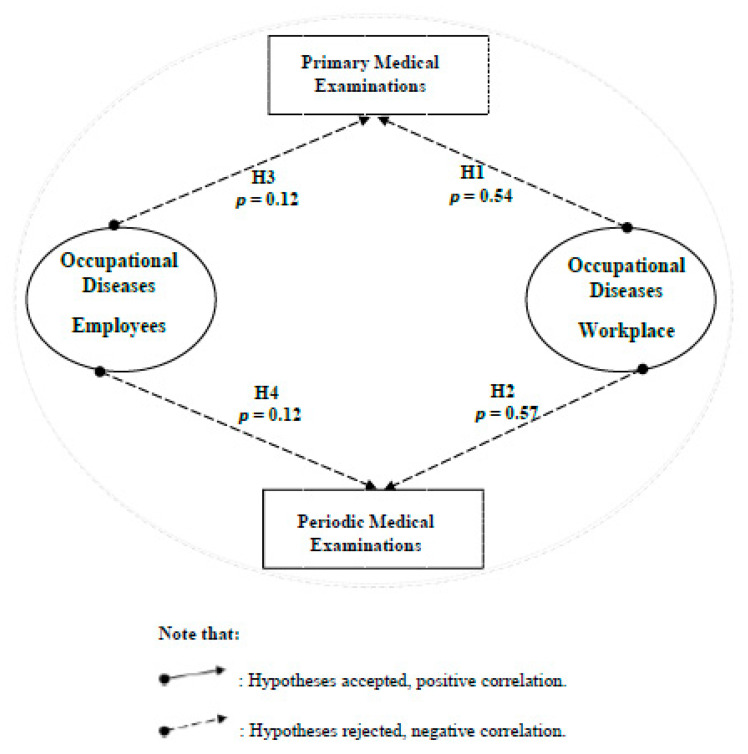
The presentation of case correlation (4) and its hypothesis testing.

**Figure 24 ijerph-18-01338-f024:**
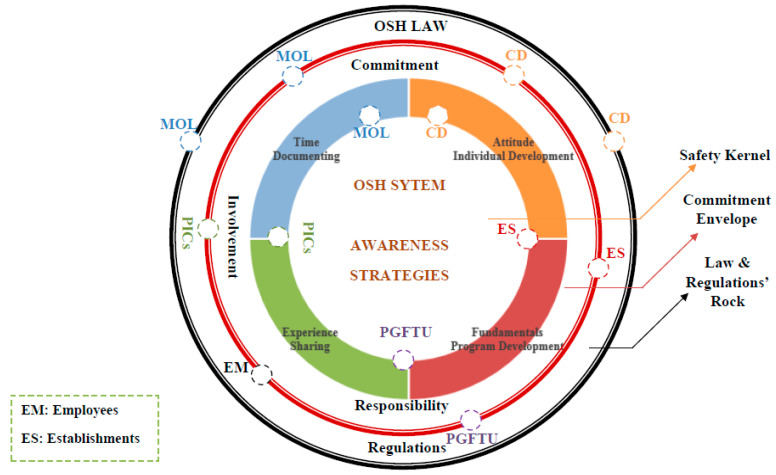
The layers and responsibilities in the safety system.

**Figure 25 ijerph-18-01338-f025:**
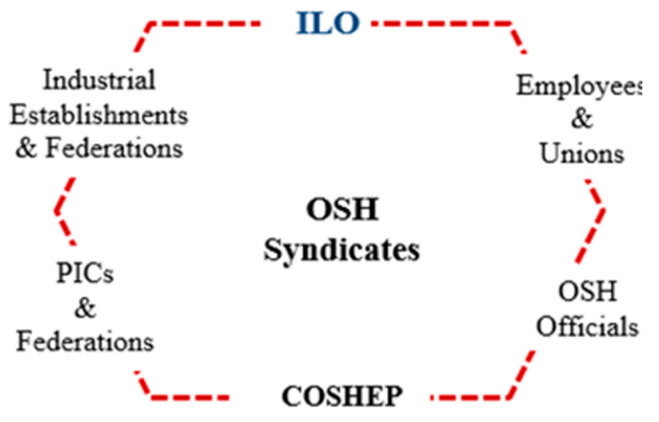
The structure of the proposed OSH syndicates, where: COSHEP = The Palestinian National Center of Occupational Safety, Health, and Environmental Protection, PICs = Palestinian Insurance Companies, ILO = International Labor Organization.

**Figure 26 ijerph-18-01338-f026:**
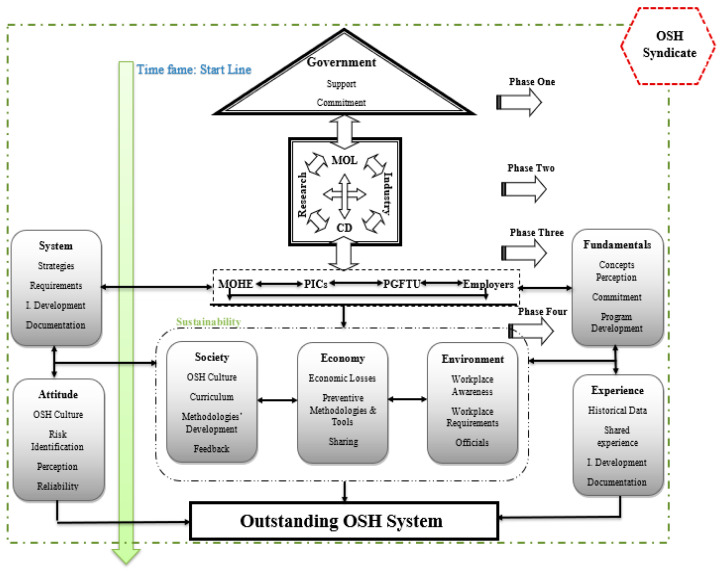
OSH outstanding conceptual framework.

**Table 1 ijerph-18-01338-t001:** The main Occupational Safety and Health (OSH) statistics between the years 2009 and 2016 according to the Ministry of Labor (MoL).

	All O.A	M.A	P.D	O.F	O.D	OWD
2009	444	270	46	9	1	105
2010	549	234	54	14	2	107
2011	399	175	31	13	3	96
2012	715	268	64	12	NA	NA
2013	752	269	20	20	NA	NA
2014	650	278	79	12	1	245
2015	664	260	35	21	NA	138
2016	682	233	20	15	NA	NA
2017	496	227	23	7	NA	320
2018	561	312	20	NA	NA	290
Total	5224	2122	368	119	7	862

Note: O.A = Occupational Accidents, M.A = Manufacturing Accidents, P.D = Permanent Disabilities, O.F = Occupational Fatalities, O.D = Occupational Diseases, OWD = Out of Work Days, NA = No clear statistics have been received in the reports.

**Table 2 ijerph-18-01338-t002:** The applied OSH rate formulas and their definitions.

OSH Rate	Symbol	Definition	Reference
**US-OSHA Rates (based on 200,000 h of worker exposure to a hazard)**			
Incident Rate	IR1	Number of occupational injuries and/or illnesses or lost workdays per 100 full-time employees.	[54]
	IR2		
Fatal Accident Rate	FAR	Number of fatalities based on 1000 employees working their entire lifetime or 10^8^ working hours over total hours worked by all employees during the period covered	[54]
Lost Time Case Rate	LTC	Number of lost time cases × 200,000 over the number of employees’ labor hour worked days.	[55]
Severity Rate	SR	Total number of lost workdays by the total number of recordable incidents.	[57]
**EU-OSHA Rates (based on 100,000 workers exposed to risk employees)**			
Fatality Rate	FR	Number of fatalities expected per person per year.	[56]
Workplace Injury Rate	IR	Number of workplace injuries of the total number of employees per 100,000 employed persons.	[57,58]
Accident Frequency Rate	AFR	Number of workplace accidents reported per number of human-hours worked × 1,000,000	[57,58]
Accident Severity Rate	ASR	Number of reported human-days lost × 1,000,000/number of human-hours worked.	[57,58]
Occupational Disease Incidents Rate	ODIR	Number of workplace diseases of the total number of employees per 100,000 employed persons.	[57]
**Severity Indicators**			
Safety Performance Factor	SPF	Number of working hours/days lost due to each occupational accident, regardless of its severity.	[58]
Frequency Severity Index	FSI	A combined formula for both AFR and ASR that gives a combined effect of accidents/injuries happened and the corresponding working days lost.	[56]

**Table 3 ijerph-18-01338-t003:** The estimated percentages of the availability of the Basic Health and Safety Requirements (BHSR) in the industrial establishments according to the 2018 workplace questionnaire analysis.

BHSR	2018	
	Workplace	Employees
Fire extinguishing means	68.0%	85.1%
First aid tools	97.7%	94.9%
Emergency exits	82.8%	77.6%
Awareness and guidance	73.0%	83.0%
Primary medical examinations	32.4%	33.5%
Periodic medical examinations	31.4%	25.8%
Reporting accidents	85.3%	90.4% *
Workers insurance	95.1%	92.9%

*** This percentage refers to the employees’ knowledge about reporting accidents and the entity they should report the accident to, not for the number of reported accidents.

**Table 4 ijerph-18-01338-t004:** Percentage of participating inspectors in the questionnaire from different official entities in charge of OSH inspection in Palestine.

	Employers	Employees
Ministry of Labor	29.5%	25.1%
Ministry of Health	13.0%	17.7%
Civil Defense	46.7%	30.2%
Ministry of National Economy	2.2%	14.3%
Ministry of Environment	1.4%	12.7%
Others	7.2%	0.0%

**Table 5 ijerph-18-01338-t005:** Number of injuries by manufacturing industries as reported by the Ministry of Labor (MoL) and the Palestinian insurance companies (PICs) during the target years.

	2009	201 0	2011	2012	2013	2014	2015	2016	2017 *
**MoL**	270	234	175	268	269	278	259	232	135
**PICs**	NA	NA	+1900	+2000	NA	NA	3500	3000	NA

* First Half of the Year. Note: NA = no data is available.

**Table 6 ijerph-18-01338-t006:** The OSH statistics of each industrial sector between the years 2009 and 2016.

Industrial Sector	Fixed Employees	O. A	O. I	O. F	O. D	LWH (days)	Estimated Losses (US$)
Leather and Shoes Industries	2445	33	31	0	8	555	10,045
Plastic Industries	6649	1387	583	1	3	1664	6600
Paper and Cartoon Industries	1989	184	186	0	3	521	49,150
Chemical Industries	1737	51	46	0	11	194	1850
Wood and Furniture Industries	2266	138	109	0	0	907	137,800
Food and beverages Industries	6669	376	262	2	26	2041	50,190
Aluminium and Metallurgical Industries	6273	941	896	1	1	3002	66,020
Entire Industrial Sector	28,028	3110	2113	4	52	8884	321,655

Note: O. A = Occupational Accidents; O. I = Occupational Injuries; O. F = Occupational Fatalities; O. D = Occupational Diseases; LWH = Lost Working Hours.

**Table 7 ijerph-18-01338-t007:** Calculated US-OSHA rates for each industrial sector and the entire industrial sector for the period 2009–2016.

Industrial Sector	IR1	IR2	FAR	LTC	SR
Leather and Shoes Industries	1.5	20.3	0.0	1.4	13.2
Plastic Industries	18.8	21.1	6.4	8.0	1.2
Paper and Cartoon Industries	8.5	24.4	0.0	8.5	2.9
Chemical Industries	3.2	9.0	0.0	2.2	3.3
Wood and Furniture Industries	7.5	50.5	0.0	5.8	6.9
Food and beverages Industries	5.3	26.1	10.4	3.7	4.7
Aluminium andMetallurgical Industries	13.7	42.8	6.5	13.0	3.2
Entire Industrial Sector	10.3	28.1	5.4	7.1	2.8

Note: IR1 = OSHA Incident Rate (based on injuries and illnesses); IR2 = OSHA Incident Rate (based on lost workdays); FAR = Fatality Accidents Rates; LTC = Lost Time Case Rate; SR = Severity Rate.

**Table 8 ijerph-18-01338-t008:** Calculated EU-OSHA rates for each industrial sector and the entire industrial sector for 2009–2016.

Industrial Sector	IR	FR	AFR	ASR	ODIR
Leather and Shoes Industries	1271.9	0.0	6.1	101.5	320.9
Plastic Industries	8820.4	1.40 × 10^−4^	94.1	105.8	39.4
Paper and Cartoon Industries	9329.1	0.0	41.9	122.2	127.5
Chemical Industries	2646.0	0.0	13.2	44.9	572.2
Wood and Furniture Industries	6640.3	0.0	37.3	252.7	0.0
Food and beverages Industries	3799.3	2.30 × 10^−4^	24.9	130.3	348.4
Aluminium andMetallurgical Industries	14,418.8	1.43 × 10^−4^	68.3	214.0	14.3
Average Entire Industrial Sector	7656.0	1.18 × 10^−4^	50.6	140.4	169.9

Note: IR = Number of workplace injuries of the total number of employees per 100,000 employed persons; FR = Number of fatalities expected per person per year; AFR = Number of workplace accidents reported per number of human-hours worked × 1,000,000; ASR = Number of reported human-days lost × 1,000,000/number of human-hours worked; ODIR = Number of workplace diseases of the total number of employees per 100,000 employed persons.

## Data Availability

The data presented in this study are available on request from the corresponding author. The data will be publicly available in a later time and can be accessed through https://scholar.najah.edu/.

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
