# Peer review of "Development of a Conceptual Framework for Occupational Safety and Health in Palestinian Manufacturing Industries"

_ijerph, 2021, doi:10.3390/ijerph18031338_

Round 1

Reviewer 1 Report

This is indeed a well prepared manuscript. Nevertheless, a few things need to be corrected. For example:

- Instead of typing "32•30%", it should be "32.30%".

- Table 1. - 2017; H1 (first half) - this abbr. should be replaced differently from Hypothesis abbrev.

- Line 127-8: "The quantitative data from the questionnaires were analyzed using (Statistical Package for Social Sciences (SPSS) Program)" - which version?

- Line 364: Do this graphs have its names?

- For the purpose of the developing a conceptual framework for occupational safety and health I would suggest to the authors to focus their attention much more on the drawn conclusions.

In addition, ref. list could be updated by some newer refs.

Reviewer 2 Report

The abstract needs to be improved upon as it fails to clearly articulate the rationale behind the research being reported in the study. For instance, the authors indicate that the study was intent on calculating the OSH trends and rates in Palestine but failed to tell the reader why such computation was imperative? And the contributions of the study’s findings are quite vague as minimal justification is made for the choice of the manufacturing sector as the study focus and not other sectors.

The introductory section needs to be revisited as it engages in a narrow reportage of the global OSH performance whilst dwelling mostly on information from the Palestinian country context. There is need for the authors to incorporate literature relating to different country contexts to enable an effective comparison during the discussion of the study’s findings.

The use of Tables in the introductory section leaves a lot to be desired. The authors should consider creating a separate literature review section where these tables will be better suited. Also, in Line 86 of the Page 3/26, the authors mention the ranking of variables according to priority but failed to provided context relating to the parameters upon which such ranking was premised.

Similarly, on Line 104 of page 3/26, the authors highlighted the number of establishments selected to participate in the study using the Multi-Stage Stratified Sampling Technique and the Thompson formula. Yet, they failed to justify the choice of these sampling techniques from a plethora of alternative sampling approaches.

The use of the term ‘addicted’ in the study is faulty and should be addressed. Questionnaires are not addicted but rather administered. This serves as an indication that the manuscript needs to be thoroughly proof-read for grammatical and syntax errors which are replete in the manuscript.

Although the presentation of the findings can be considered as in-depth and well-articulated, it is noted that the methods section did not properly describe the methods that were deployed for data collection and analysis. This should be elucidated subsequently.

The discussion of findings section fails to engender a comparison between the study’s findings and the findings of similar studies focusing either on different country contexts or on different economic sector contexts within Palestine.

There is need for a distinct conclusion section as this is missing in the manuscript.

Reviewer 3 Report

The manuscript presents an analysis of OSH conditions in Palestine and proceeds from there to the conceptual presentation/proposal of a structure for the development in several stages of a safety system.

It is an interesting work and is within the scope of the journal. There are, however, questions that must be answered before it can be considered for publication.

Keywords - two of the keywords do not occur in the text (Industry's Severity, Outstanding Framework). They should be referred to in the text or replaced by others more integrated with the developed work.

Referencias: – As referências mais recentes são de 2017. Devem ser procuradas outras mais recentes que permitam ter uma visão mais atualizada da investigação na área, nomeadamente com experiências do mesmo tipo em outros outros países. Essa comparação poderia dar outra dimensão ao trabalho desenvolvido.

Lines 30-41 - The statement seems to contain an analysis bias and needs better reasoning. From the presented references, this conclusion cannot be drawn. The increasing percentage in accidents due to human error may be due, simply because errors due to other causes have decreased, without the corresponding decrease in errors due to human factors.

Line 61 (figure 1) - It is necessary to clarify in the text the difference between Occupational accidents and Manufacturing accidents in the context of the manuscript to be clear what each term means.

Line 82 (Table 1) - The work justification would be more robust if data related to the various sources were presented. According to what has been said, there are significant discrepancies between them.

Line 82 (Table 1) - The designation of the variables would be better if placed as a note at the bottom of the page. Check all tables.

Line 96 - Acronyms - All acronyms must be defined the first time they occur in the text (USAID, PFI, check for others);

Line 117 - It would be convenient to prove that passing from 265 to 102 questionnaires did not influence the results' representativeness, according to the Multi-Stage Stratified Sampling Technique and Thompson formula.

Line 172 (results) - The results of the different parameters analyzed are presented in tables and graphs, the latter being of various types. The presentation of data must be homogenized coherently throughout the work.

Line 248 - Throughout the text, the words "employers" and "establishments" appear with meanings, sometimes apparently identical. Please clarify.

Line 274 - The legend colours are barely noticeable.

Lines 296-297 - It would also be interesting to have other more recent studies.

Lines 298-299 - It should be specified that it is in Palestine.

Line 303 - Figure 11 - The YY axis on the left side has no legend.

Line 319 - HSE 44 - Place at least one link for this data.

Line 364 - Figure 15 - The amplitude of the scales on the YY axis in the different graphs in the figure must be the same, to have some comparability between them.

Line 383 - Figure 17 - all formulas used must be presented. Check throughout the text.

Line 405 - Figure 19 - What is the real meaning of "Industrial Sector"? It appears in the chart legend and as a label for the last column of the chart. Could the latter be "Other Industrial Sectors"?

Line 421 - Figure 20 - LWD / Hs is not defined in the text.

Line 247 - Delete parentheses.

Line 248 - Delete parentheses

Line 448 - "I It" - delete "I".

Line 507 - Place arrows on all lines so that the flows can be understood.

Round 2

Reviewer 1 Report

Something is wrong with the document I have downloaded.

After adjusting it, I would suggest the manuscript for publishing.

Author Response

We thank you very much for your instructive comments. 

Reviewer 2 Report

The suggestions made in the first review have now been incorporated in the manuscript. Accordingly, the manuscript better conveys its message in its current form. 

Author Response

(The authors gave the same response as above.)

Reviewer 3 Report

The performed review responded to the request. Only minor checks are necessary.

The formatting needs to be homogenized throughout the text according to the IJERPH template.

- Sometimes is written MOL other MoL please always use the same notation.

- Sometimes is written MOH other MoH please always use the same notation.

- Subchapter 2.3 has only two lines, so it should be integrated into one of the other subchapters.

Lines 58-59 – where is “Table (1)” must be “Table 1”.

Line 575 – where is Figure (25) and Figure (26), must be Figure 25 and Figure 26

Line 186: (Figure 1&2); Line 527: (Figures 25 and 26). Please always use the same notation according to the template.

Line 212 - Please check the positioning of Figure 2 and the respective caption.

Line 254 – delete “below”

Table 4 - check the sum of the “Employers” column. The sum is different from 100%.

Line 340 - “Table 5” appears twice.

Table 8 - must be called from the text before it occurs.

Lines 408-409 - The graphs must be homogenized according to the following ones.

Line 556 - the caption is not in the right place.

Author Response

We thank you very much for your instructive comments. Below is our response:

The performed review responded to the request. Only minor checks are necessary.

The manuscript was thoroughly checked and amended, and all comments were addressed.

The formatting needs to be homogenized throughout the text according to the IJERPH template.

Many thanks. We have checked the formatting and homogenized it.

- Sometimes is written MOL other MoL please always use the same notation.

MoL notation was used throughout the manuscript and changed in Figures 1 and 2 as well.

- Sometimes is written MOH other MoH please always use the same notation.

MoH notation was used throughout the manuscript

- Subchapter 2.3 has only two lines, so it should be integrated into one of the other subchapters.

Section 2.3 (OSH framework) was combined with section 2.2.4, and also noted at the end of the introduction

Lines 58-59 – where is “Table (1)” must be “Table 1”.

Done

Line 575 – where is Figure (25) and Figure (26), must be Figure 25 and Figure 26

Done

Line 186: (Figure 1&2); Line 527: (Figures 25 and 26). Please always use the same notation according to the template.

Done

Line 212 - Please check the positioning of Figure 2 and the respective caption.

Done

Line 254 – delete “below”

Done

Table 4 - check the sum of the “Employers” column. The sum is different from 100%.

Done, the sum is 100%

Line 340 - “Table 5” appears twice.

Done

Table 8 - must be called from the text before it occurs.

The statement “The average IR value presented in Table 8 and Figure 12 rings an alarm for Palestine's severe safety situation compared to other countries'‎‎‎‎‎‎‎‎‎‎‎‎ IR values.” was moved ahead to Table 8

Lines 408-409 - The graphs must be homogenized according to the following ones.

Done

Line 556 - the caption is not in the right place.

All captions were amended.